# How to control the learning rate of adaptive sampling schemes

**Lorenz Berger**[1,2], **Eoin Hyde**[1,2], **Nevil Pavithran**[3],
**Faiz Mumtaz**[3], **Felix Bragman**[2], **M. Jorge Cardoso**[2], **Sébastien Ourselin**[2]
[1]Innersight Labs, London UK, [2]WEISS, University College London, [3]Royal Free Hospital, London UK,
`lorenz@innersightlabs.com`

## Abstract

Deep convolutional neural networks have shown excellent performance in object recognition tasks and dense classification problems such as semantic segmentation. However training deep neural networks is still challenging and can require large amounts of computational resources to find network hyperparameters that result in good generalization properties. This procedure can be further complicated when an adaptive/boosted/online-hard-mining sampling scheme is used which varies the amount of information in mini-batches throughout training. In this work we address the task of tuning the learning rate schedule for Stochastic Gradient Descent (SGD) whilst employing an adaptive sampling procedure. We review recent theory of SGD training dynamics to help interpret our experimental findings, give a detailed description of the proposed algorithm for optimizing the SGD learning rate schedule and show that our method generalizes well and is able to attain state-of-art results on the VISCERAL Anatomy benchmark.

## 1 Introduction

To address the problem of efficiently training convolutional neural networks (CNNs) on large and imbalanced datasets. Training strategies that dynamically sample the training data, by evaluating the classification error throughout training to effectively speed up training and avoid over-sampling data that contains little extra information, have been proposed [1, 2, 3]. This prevents the vast number of easy negatives from overwhelming the optimization process during training. However they introduce an additional non-linearity into the stochastic gradient descent process (SGD) by changing how data samples are chosen throughout training, by making the sampling dependent on current model performance. As a result, adaptive sampling can even cause the loss of minibatches to increase during training, since more difficult samples are preferred [2]. This makes it challenging to choose an effective learning rate schedule that adapts to the ever changing amount of information/loss in the updates during training.

Hyperparameter tuning is arguably the most time-consuming part of deep learning, with thousands of productive hours sacrificed and many papers outlining best tuning practices [4, 5, 6, 7]. State-of-the-art optimizers, such as Adagrad [8], RMSProp [9] and Adam [10], make things easier by adaptively tuning the learning rate individually for each variable. However, recent theory and experiments outlined in [11], show that hand-tuned SGD achieves superior results, and finds minima that generalize well [12, 11]. Naive methods for tuning the learning rate schedule of SGD, like grid-search, are prohibitively expensive for all but the smallest problems. Therefore hand-tuning of the hyperparameters such as the initial learning rate and learning rate decay (when to reduce the learning rate and by how much) is recommended [11]. Moreover, for applications using deep learning for medical image segmentation, it is often observed by practitioners that a good choice of these hyperparameters, can make the difference between a bad and a good segmentation algorithm [13].

This paper addresses the problem of efficiently training a Convolutional Neural Network architecture that uses an adaptive sampling strategy. We make the following contributions:

1st Conference on Medical Imaging with Deep Learning (MIDL 2018), Amsterdam, The Netherlands.

- We review available theory of SGD training dynamics to gain insights into how to best adapt the learning rate throughout training.

- We propose a simple algorithm to adaptively tune the learning rate, set within the framework of genetic algorithms, and thus are able to effectively and automatically tune the amount of information during minibatch updates resulting in competitive performance compared with hand-tuned SGD.

- Provide experimental results that support the theory, validate the proposed algorithm and give state-of-the-art segmentation results on the VISCERAL anatomy benchmark.

## 2 Background

### 2.1 Stochastic Gradient Descent

Stochastic gradient descent (SGD) remains the dominant optimization algorithm of deep learning. Many recent works have interpreted SGD as a stochastic differential equation [15, 16, 17, 18, 19] and asymptotic theory for stochastic gradient descent has been developed [20]. Following [19, 18], the stochastic gradient update, at time $t$, can be written as,

$$\boldsymbol{\theta}(t + 1) = \boldsymbol{\theta}(t) - \eta \boldsymbol{g}_S(\boldsymbol{\theta}(t)), \tag{1}$$

where $\boldsymbol{\theta}$ are the model parameters (weights), $\eta$ is the learning rate and $\boldsymbol{g}_S$ is the gradient of the loss function of a minibatch of size $S$, given by $\boldsymbol{g}_S(\boldsymbol{\theta}) = \frac{\partial}{\partial \boldsymbol{\theta}} \frac{1}{S} \sum_{\boldsymbol{x}_n \in B} l(\boldsymbol{\theta}, \boldsymbol{x}_n)$. By the central limit theorem, assuming the noise in the stochastic gradient is Gaussian with covariance matrix $\frac{1}{S} \boldsymbol{C}(\boldsymbol{\theta})$,

$$\boldsymbol{g}_S(\boldsymbol{\theta}) = \boldsymbol{g}(\boldsymbol{\theta}) + \frac{1}{\sqrt{S}} \Delta \boldsymbol{g}(\boldsymbol{\theta}), \text{ where } \Delta \boldsymbol{g}(\boldsymbol{\theta}) \sim N(0, \boldsymbol{C}(\boldsymbol{\theta})). \tag{2}$$

Combining (1) and (2), and assuming discretisation errors are small this leads to the following continuous-time stochastic differential equation,

$$d\boldsymbol{\theta}(t) = -\eta \boldsymbol{g}(\boldsymbol{\theta}) dt + \frac{\eta}{\sqrt{S}} \boldsymbol{B}(\boldsymbol{\theta}) d\boldsymbol{W}(t), \text{ where } \boldsymbol{W}(t) \sim N(0, \boldsymbol{I}), \tag{3}$$

where the property $\boldsymbol{C}(\boldsymbol{\theta}) = \boldsymbol{B}(\boldsymbol{\theta}) \boldsymbol{B}(\boldsymbol{\theta})^T$ has been applied, and it is assumed that the loss function is well approximated locally by a quadratic, $L(\boldsymbol{\theta}) = \frac{1}{2} \boldsymbol{\theta}^T \boldsymbol{A} \boldsymbol{\theta}$, where $\boldsymbol{A}$ is the symmetric Hessian matrix. The SDE (3) is known as a Ornstein-Uhlenbeck process [21] which has been well studied [22]. Note that larger learning rates, $\eta$, in (3) result in a larger drift term and hence faster initial descent along the gradient of the loss. However, the same factor is also multiplied to the noise term, causing greater asymptotic fluctuations.

Now assuming isotropic covariance $\boldsymbol{C}(\boldsymbol{\theta}) = \sigma^2 \boldsymbol{I}$, where $\sigma$ is a constant, and defining $P(\boldsymbol{\theta}, t_\eta)$ to be the probability density of $\boldsymbol{\theta}$ at time $t$, we arrive at the Focker-Plank equation [18, 22],

$$\frac{\partial P(\boldsymbol{\theta}, t)}{\partial t} = \nabla_{\boldsymbol{\theta}} \cdot [\eta \boldsymbol{g}(\boldsymbol{\theta}) P(\boldsymbol{\theta}, t)] + \frac{\eta^2 \sigma^2}{2S} \nabla_{\boldsymbol{\theta}}^2 P(\boldsymbol{\theta}, t). \tag{4}$$

After rescaling time to be $t_\eta = t\eta$ we get,

$$\frac{\partial P(\boldsymbol{\theta}, t_\eta)}{\partial t_\eta} = \nabla_{\boldsymbol{\theta}} \cdot [\boldsymbol{g}(\boldsymbol{\theta}) P(\boldsymbol{\theta}, t_\eta)] + \frac{\eta \sigma^2}{2S} \nabla_{\boldsymbol{\theta}}^2 P(\boldsymbol{\theta}, t_\eta), \tag{5}$$

Now the ratio $\frac{\eta \sigma^2}{2S}$, is a clear measure of the amount of diffusivity or noise in the system. If we now run SGD, asymptotically modeled by (5), for long enough until we reach an equilibrium described by a Gaussian distribution [22]. Following [18] and assuming that the loss is locally strictly convex with Hessian $\mathbf{H}_A$ and loss $L(\boldsymbol{\theta}_A)$ at a minimum $\boldsymbol{\theta}_A$, then the probability of ending in a region near minima $\boldsymbol{\theta}_A$, can be expressed as,

$$P(\boldsymbol{\theta}_A) \propto \frac{1}{\sqrt{\det \mathbf{H}_A}} \exp \left( -\frac{2L(\boldsymbol{\theta}_A)}{n\sigma^2} \right), \tag{6}$$

where $n = \frac{\eta}{S}$ is the amount of noise during SGD. The above result shows that the probability of landing in a specific minimum depends on the learning rate, batch-size, covariances of the gradients

and loss of the minimum. The fact that the loss is divided by $n$ in equation 6 emphasizes their direct relationship, and shows that the higher the noise ratio $n$, which we can control directly, the less granular the loss surface appears to SGD. The determinant of the Hessian $\det\mathbf{H}_A$ can be interpreted to describe the shape of the converged local minima, a smaller determinant implying a wider minima with better generalisation properties. Note that in the above section several assumptions were made to derive a closed form solution for the equilibrium distribution of SGD. Many of these, such as the linearity of the covariances will not hold in practice, see [19, 18] for discussions, and further theoretical analysis is needed.

Based on similar insights, several strategies for setting and decaying the learning rate throughout training have been proposed, to optimize for flat minima that result in good generalisation, and fast training. In [23] the linear noise scaling rule $n = \frac{\eta}{S}$ is also observed and exploited to train ResNet-50 on ImageNet in one hour, by parallelizing the training on large batches and reducing the learning rate by $1/10$, after $30, 60$ and $80$ epochs. In [15] the linear scaling rule was used to show that the same learning curve on both training and test sets can be achieved by increasing the batch size during training instead of decreasing the learning rate, to allow for further parallelization of large batch training. Applying optimal control theory to minimize the loss from equation (3), and assuming that the loss is quadratic, in [17], they obtain a learning rate annealing schedule which says that the maximum learning rate should be used for descent phases, whereas $\sim 1/t$ decay on learning rate should be applied after the onset of fluctuations, during the second phase of SGD. Authors in [24] stress the importance of having the highest possible diffusion rates (which do not result in numerical instability) and a large number of training iterations at the start of training to reach wide ('flat') minima that result in good generalisation. Another approach [25] eliminates the need to experimentally find the best values and schedule for the learning rates by monotonically decreasing the learning rate, and letting the learning rate cyclically vary between reasonable boundary values [25].

Due to the infancy of the described theoretical tools available and automated learning rate schedulers, practitioners still spend vast resources on tuning these critical hyper-parameters. For example in medical imaging, to achieve state-of-the-art results in brain lesion segmentation, [7] had to identify an optimal predefined learning rate schedule, halving the learning rate after epoch $12, 16, 19, 22, 25, 28, 31, 34, 37, 40, 43$ and $46$. To overcome these challenges we propose an adaptive and simple to implement method to optimally tune the learning rate throughout training.

## 3 Methods

### 3.1 Adaptive learning rate scheduling

We evolve the learning rate throughout training, by evaluating the CNN's validation performance on a small population of concurrent CNN runs, each trained with a different learning rate, for some validation period. The weights of the best performing run at the end of the period are then used to spawn a new set (population) of CNN runs, each with a different learning rate. This simple algorithm can also be extended to any other hyperparameter. This simple procedure is outlined in more detail in Algorithm 1. The number of parallel runs used can be chosen arbitrarily. However a clear disadvantage, as with any genetic type algorithm, is that the more parallel runs are spawned, the more parallel processing is required. The described algorithm can easily be generalised to fit within the framework of evolutionary algorithms and therefore can be extened to deal with more complex genetic operators and fitness functions, depending on the application.

### 3.2 isample: Adaptive sampling strategy

To overcome issues of inefficient training on sparse datasets such as CT data we use the previously described isample algorithm [2]. The sampling algorithm prioritises sampling from training data that at the time of training produces large training error. This affects the loss and noise in the minibatches selected throughout training, and can even cause the loss function to increase, as more difficult samples, which have a high loss, are chosen by the sampler. Therefore introducing a further non-linearity into the training process, and motivating the need for an automated learning rate scheduler.

This simple method is described in Algorithm 2, where $\mathcal{U}(0, 1)$ is a random number drawn from the uniform distribution and $\boldsymbol{E}_i$ refers to the error map of the $i^{th}$ training image. Error maps can easily

---
**Algorithm 1** AutoLR: Adaptive learning rate scheduler
---
Choose, $\beta$ the step length, the number of runs, $R_i$, and the initial learning rate and modulating factor for each run $\eta_i, \lambda_i$.
**while** CNN training **do**
    **while** training each run, $R_i$ in parallel, for $\gamma$ epochs **do**
        Record validation performance of each run, $R_i$, at each epoch in the vector $\boldsymbol{v}_i$ of length $\beta$.
    **end while**
    Identify the best performing run, $R_i^* = \max\limits_{R_i} \{\max[\boldsymbol{v}_i]\}$

    Set the optimal learning rate to be the learning rate of the best performing run, $\eta^* = R_i^*, \eta$
    Update the learning rate of each run, $\eta_i = \eta^* \lambda_i$
**end while**
---

---
**Algorithm 2** isample: adaptive sampling algorithm
---
Initialise error maps for every image in the training data: $\boldsymbol{E}_i(x) = 1$.
**while** CNN training **do**
    **while** training for 1 epoch **do**
        **while** filling batch with patches **do**
            Pick an image $\boldsymbol{I}_j$ from the training set $\boldsymbol{I}^*$.
            Pick a class $k$ from the corresponding label map $\boldsymbol{L}_j$.
            Pick a patch in image $\boldsymbol{I}_j$, centered at location $\boldsymbol{c}$, where $\boldsymbol{L}_j(\boldsymbol{c}) = k$.
            Accept patch into batch if $\boldsymbol{E}_i(\boldsymbol{c}) > \mathcal{U}(0,1)$.
        **end while**
        Back-propagate loss of batch and update the current CNN weights: $\boldsymbol{w}$.
    **end while**
    Select a subset of images, $\boldsymbol{I}^*$, and label maps, $\boldsymbol{L}^*$, from the training set:
    **for** $[\boldsymbol{I}_k, \boldsymbol{L}_k] \in [\boldsymbol{I}^*, \boldsymbol{L}^*]$ **do**
        Update error maps: $\boldsymbol{E}_k(\boldsymbol{x}) = 1 - \text{CNN}(\boldsymbol{w}, \boldsymbol{I}_k(\boldsymbol{x}))_{\boldsymbol{L}_k(\boldsymbol{x})}$
    **end for**
**end while**
---

be calculated, either after each epoch or concurrently to the training process, as

$$\boldsymbol{E}_k(\boldsymbol{x}) = 1 - \text{CNN}(\boldsymbol{w}, \boldsymbol{I}_k(\boldsymbol{x}))_{\boldsymbol{L}_k(\boldsymbol{x})}, \tag{7}$$

where $\text{CNN}(w, \boldsymbol{I}_k(\boldsymbol{x}))_{\boldsymbol{L}_k(\boldsymbol{x})}$ is a map of the CNN predictions over the full training image $\boldsymbol{I}_k$, evaluated using the most current weights, $\boldsymbol{w}$, and outputting the probability of the true class label $\boldsymbol{L}_k(\boldsymbol{x})$, at position $\boldsymbol{x}$.

## 3.3 Neural Network Architecture

For the dual path network architecture we build on several previous ideas [7, 26, 27]. Compared to the 3D network outlined in [7], we further develop the architecture by replacing the standard convolution layers with popular resnet blocks [28], and increase the maximum network depth from 11 layers in [7] to 21 layers. By having a deeper network and a down sampled pathway with input resolution $1/4$ of the original resolution, we obtain a large receptive field of size $123^3$ whilst maintaining a deep high resolution pathway that does not compromise the resolution through pooling layers. The architecture results in a total of 649,251 parameters.

A sketch of the architecture is given in Figure 1a. In Figure 1a, numbers inside round brackets give the input dimensions of each block. For the training stage these dimensions have been chosen carefully to balance memory usage with processing speed. During testing the dimensions may be chosen as large as can be fit into memory, to take advantage of the fully convolutional inference. Numbers in square brackets refer to the number of feature maps used at each layer. The proposed configuration allows for a large number of samples ($3D$ patches) per batch to ensure balanced class sampling and effective optimization, whilst maintaining a deep and wide enough network to capture the high variability and spatial semantics of the data.

The blocks labeled 'Conv' are standard convolutional layers with kernel size $3 \times 3 \times 3$. The blocks labeled 'RA_block' and 'RB_block' are standard and bottleneck resnet blocks, respectively, as detailed in [28]. Each fully connected layer is preceded by a dropout layer with probability $0.5$, and a softmax nonlinearity is used as a final classification layer.

The rationale for having a deep low resolution path is to further increase the receptive field and allow for complex higher level features to be learned i.e where an organ is positioned in relation to other structures. To minimize the memory footprint, the high definition path is chosen to be slightly shorter than the low resolution path. This seems reasonable as this path should learn texture information which is likely to require fewer layers and non-linearites. Further details on training hyper-parameters are given in section 4.1.

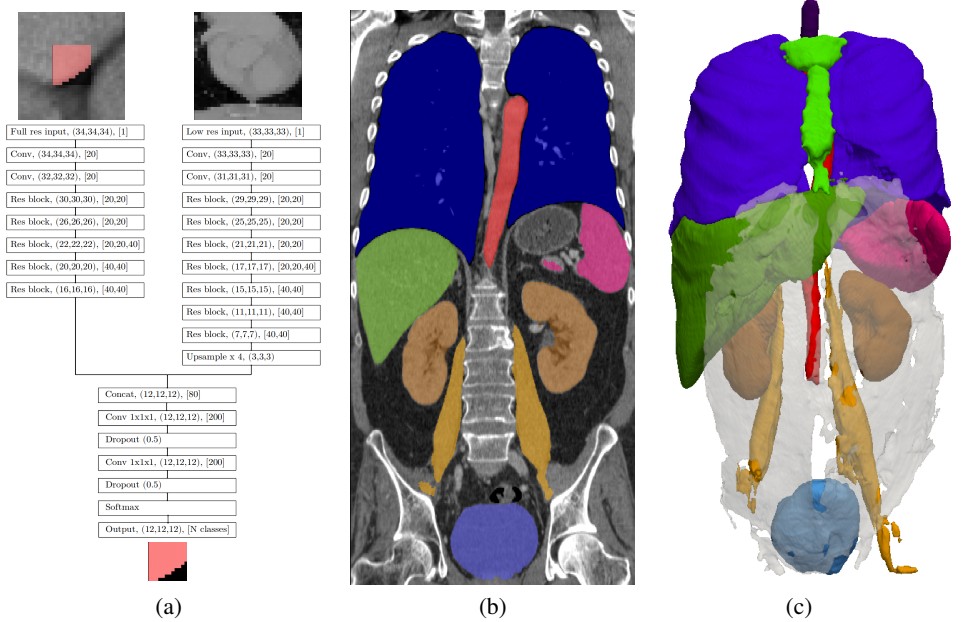

(a)  (b)  (c)

Figure 1: (a) The proposed dual path CNN architecture. (b) Coronal slice of a CT scan with overlaid segmentation output, described in section 4.2. The organs visible in this slice are: lungs (green), liver (red), spleen (light blue), psoas major muscle (dark blue), kidneys (brown) and bladder (yellow). (c) 3D surface rendering of the segmentation.

## 4  Experiments

We trained, validated and tested the automatic segmentation method on contrast enhanced CT scans from the VISCERAL Anatomy 3 dataset, made up of 20 scans [29]. The scans are form a heterogeneous dataset with various topological changes between patients, and manual segmentations are available for a number of different anatomical structures. We randomly split the training set into 14 scans for training (70%), 2 scans for validation (10%) and 4 scans for hold-out testing (20%).

### 4.1  CNN training setup

During training we perform data augmentation by re-sampling the 3D patches to a $[1mm, 1mm, 1.5mm] + \mathcal{U}(-0.1, 0.1)$ resolution. We also rotate each patch by $[\mathcal{U}(-10, 10), \mathcal{U}(-4, 4), \mathcal{U}(-4, 4)]$ degrees. We set voxels with values greater than 1000 to 1000, and values less than $-1000$ to $-1000$, and divide all values by a constant factor of 218 (the standard deviation of the dataset). We use Glorot initializations [30] on all convolution layers. For batchnorm layers we use the initializations technique described in [23]. We impose $L_2$ weight decay of size 0.0001, on all convolutional layers except on the last fully convolutional layer before the final softmax non-linearity. Using techniques described in [23] we make use of large batch sizes and large learning rates. We use SGD with Nestrov momentum set at 0.8 [31], and each batch contains 40 patches, sampled randomly from the whole training set. We run each epoch for 100 batches. We also employ a linear learning rate warm up schedule as described in [23] for 10 epochs when starting to train or switching between learning rates. This is especially important

when increasing the learning rate, to avoid gradient blow up and accumulation due to momentum, as described in [23].

For Algorithm 1 we chose to use three runs with the following configurations, $R_0 : \eta_0 = 0.05, \lambda_0 = 2$, $R_1 : \eta_1 = 0.01, \lambda_1 = 1$ and $R_2 : \eta_2 = 0.005, \lambda_2 = 0.5$. We found that having one run that explores at a higher learning rate ($R_0$), one continuing at the same as the previously used learning rate ($R_1$), and one that explores at half the learning rate ($R_2$) worked well, and allowed the algorithm to effectively adapt to a well performing learning rate schedule. This configuration was used for all subsequent experiments, unless stated otherwise.

## 4.2 Multi-organ segmentation on the VISCERAL CT dataset

We now present results of experiments performed using the multi-organ VISCERAL anatomy CT-enhanced dataset.

Figure 2 demonstrates how the algorithm adapts to find an optimal learning rate schedule. During this example we only show the first 200 epochs of training to best visualise the method's dynamics. Figure 2b shows the learning rate throughout training by the three spawned runs. After training for 50 epochs, the recorded validation scores are evaluated, which for these experiments have been chosen to be the mean dice score of all the organs segmented on the full validation scans. In Figure 2a it can be seen that the maximum dice score for this first period was achieved by the run colored in red, training at a learning rate of $\eta = 0.05$, such that $\eta* = 0.05$ in Algorithm 1. Therefore after the first step period, following epoch 50, the three runs evolve such that $\eta_0 = \eta^* \lambda_0 = 0.05 \times 2 = 0.1$, $\eta_1 = 0.05 \times 1 = 0.05$ and $\eta_2 = 0.05 \times 0.5 = 0.025$ throughout the next step period. From Figure 2b we can see that this causes the learning rate to initially increase, which speeds up training and allows SGD to move away from its initialisation, and later reduce the learning rate. The loss of the three runs is shown in 2c. The higher the learning rate the more stochasticity can be seen in the SGD loss function, which agrees with equation 5.

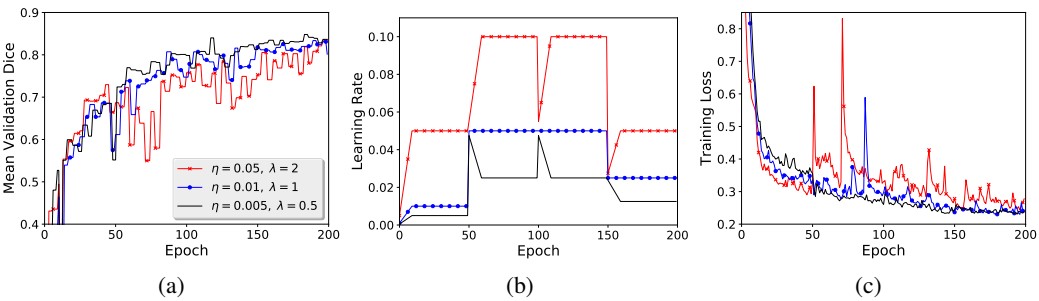

Figure 2: (a) Mean organ validation dice score on validation scans. (b) Learning rate throughout training of three parallel runs during AutoLR. (c) Cross entropy loss during training.

To benchmark the proposed learning rate scheduling Algorithm 1 we select several handcrafted learning rate training schedules and compare their performance against the proposed method. Before arriving at these handcrafted many initial experiments (not shown here) were run to get close to the optimal initial learning rates and decay schedules shown in in Figure 3. To produce the results shown in Figure 3, five weeks of training time using eight NVIDIA Tesla K40s was required.

With and without adaptive sampling used (algorithm 2) we find that AutoLR performs competitively with the top hand tuned learning rate schedules, shown in Figure 3a and 3c, and the automatically found learning rate schedule follows a similar path to the top performing schedule, shown in Figure 3b and 3d.

| Dataset (training updates) | Sched A | Sched B | AutoLR | Sched A + is | Sched B + is | AutoLR + is |
|---|---|---|---|---|---|---|
| Validation ($25k$) | 0.773 | 0.788 | 0.827 | 0.792 | 0.839 | 0.866 |
| Validation ($100k$) | 0.859 | 0.866 | 0.848 | 0.881 | 0.867 | 0.880 |

Table 1: Mean validation dice scores after $25k$ and $100k$ SGD updates, for two different hand tuned learning rate schedules: Sched A: $\eta_0 = 0.05, \lambda = 0.1, \gamma = 400$, Sched B: $\eta_0 = 0.025, \lambda = 0.1, \gamma = 400$, and AutoLR, with and without adaptive sampling (+ is).

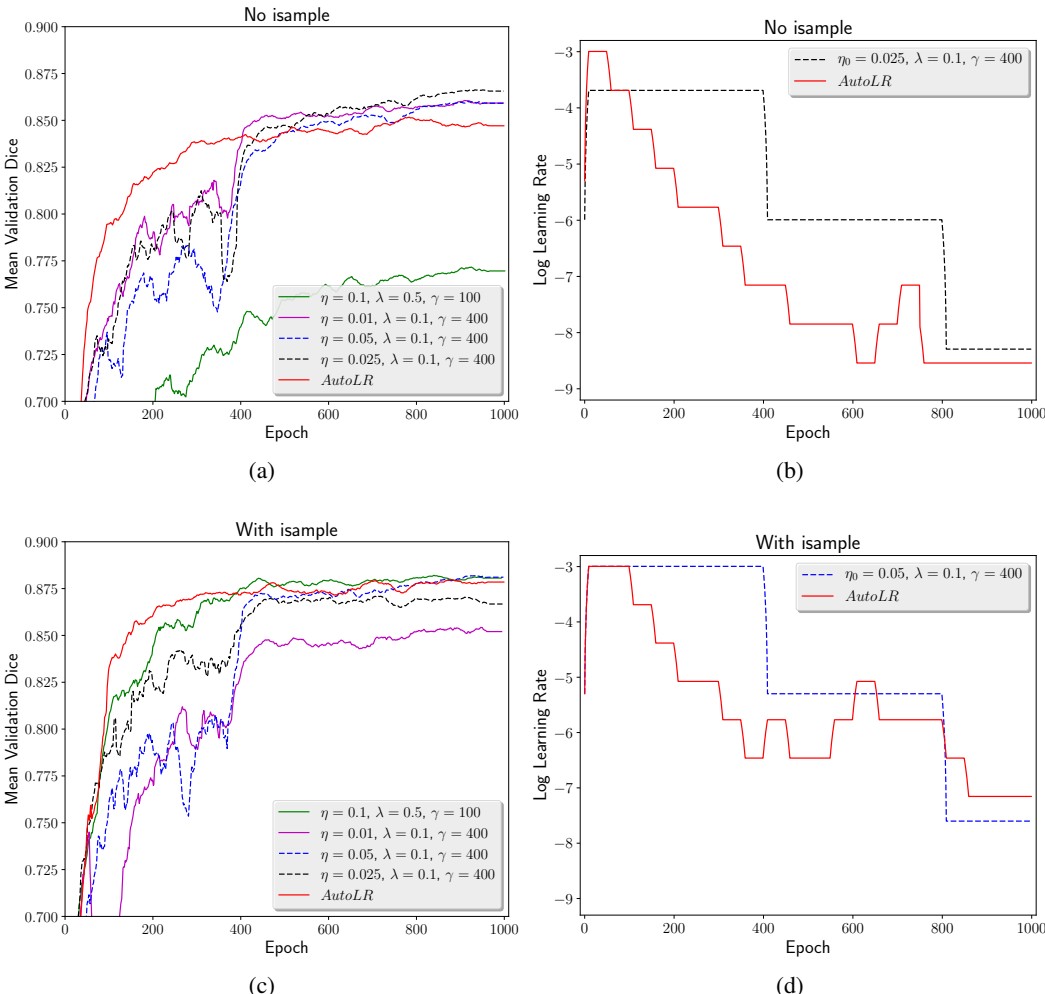

Figure 3: Mean validation dice scores throughout training of four hand tuned schedules and an AutoLR schedule, with (c) and without (a) using adaptive sampling. The learning rate of the top performing hand tuned schedule and the AutoLR learning rate schedule, with (d) and without (b) using adaptive sampling.

Table 1 shows the average validation dice scores achieved by AutoLR and the best manually tuned learning rate schedules. The top performing manually tuned learning rate schedule when training the CNN without isample is 'Sched B', with hyperparameters $\eta_0 = 0.025, \lambda = 0.1, \gamma = 400$. The top performing manually tuned learning rate schedule when training the CNN with isample is 'Sched A', with hyperparameters $\eta_0 = 0.05, \lambda = 0.1, \gamma = 400$. AutoLR is able to achieve a high dice score of 0.827 (without isample) and 0.866 (with isample) after $25k$ updates, significantly improving on both 'Sched A' and 'Sched B'. This can also be seen from the validation dice curves in Figure 3a and 3c. AutoLR achieves this by moving to a high learning rate at the start of training which speeds up

|  | Aorta | Lung | Kidney | PMajor | Liver | Abdom | Spleen | Sternum | Trachea | Bladder |
|---|---|---|---|---|---|---|---|---|---|---|
| Ga et al [32] | 0.785 | 0.963 | 0.914 | 0.813 | 0.908 | - | 0.781 | 0.635 | 0.847 | 0.683 |
| Jimenez et al [33] | 0.762 | 0.961 | 0.899 | 0.797 | 0.887 | 0.463 | 0.730 | 0.721 | 0.855 | 0.679 |
| Kéchichian et al [34] | 0.681 | 0.966 | 0.912 | 0.802 | 0.933 | 0.538 | 0.895 | 0.713 | 0.824 | 0.823 |
| Vincent et al [35] | 0.838 | 0.972 | 0.935 | 0.869 | 0.942 | - | - | - | - | - |
| AutoLR + isample | 0.875 | 0.979 | 0.922 | 0.851 | 0.929 | 0.824 | 0.895 | 0.829 | 0.910 | 0.860 |
| AutoLR + isample + post-proc | 0.887 | 0.983 | 0.924 | 0.861 | 0.953 | 0.830 | 0.947 | 0.855 | 0.927 | 0.903 |
| Inter-annotator agreement | 0.859 | 0.973 | 0.917 | 0.823 | 0.965 | 0.673 | 0.934 | 0.810 | 0.877 | 0.857 |

Table 2: Dice scores for different automatic multi-organ segmentation methods and inter-annotator agreement results [29] on the VISCERAL dataset.

overall training dynamics as can be seen from the natural rescaling of time argument, involving the learning rate, in equation 5. After $100k$ the manually tuned learning rate schedules, which train at constant learning rates for long periods before dropping, catch up and slightly outperform AutoLR. We hypothesis this is because the strategy of 'train longer, generalize better', as described in [24], allows SGD to move further away from the initialisation before converging to a minimum, resulting in wider minima that generalise better. This effect of diffusing away from minima can also be interpreted by looking at equation 5 and observing that a high and prolonged learning rate maintains stochasticity/diffusivity during the optimization process. With the adaptive sampling used, 'Sched A' ($\eta_0 = 0.05$) performs better after $100k$ updates of training than 'Sched B' ($\eta_0 = 0.025$). However when not using adaptive sampling, then 'Sched B' is better than 'Sched A'. This indicates, that when using an adaptive sampler such as isample that selects patches (samples) with higher loss and thus higher informative value, that an overall higher learning rate is favored during training, which is supported by the inverse relationship between the loss and learning rate in equation 6.

The results of our proposed method and other state-of-the-art methods on this dataset, also summarized in [29], are given in Table 2. Inference on a full size CT scans takes $\sim 65$ seconds using four Tesla K40 GPU cards, each with 4GB of RAM. We note that because the cloud-based evaluation service [29] containing the test data was closed at the time of running these experiments, we were not able to evaluate our method on the test data, thus making direct comparisons to previous methods difficult. As previously mentioned, we trained our method on $70\%$ of the data (14 scans), validated on ($10\%$) of the data (2 scans) and tested it on $20\%$ of the data (4 scans). We also include results that post-processes the output segmentation maps (maximum class probability at each voxel), by applying a simple filter that only retains the largest connected binary object for each organ, thus removing small misclassified objects.

# 5  Discussion

From the experiments of our proposed method AutoLR and hand tuned schedules, we find that AutoLR is highly competitive with fine tuned learning rate schedules that took many weeks of experimentation to arrive at. We also experimented using the adaptive optimization algorithm ADAM [10] without decaying the learning rate, but could only achieve poor generalisation performance. We also confirm previous findings that show adaptive sampling can significantly speed up training time [1, 2].

Further, in practice we found that it was very difficult and took large amounts of time and computation to design an optimal handcrafted learning rate schedule, as it is extremely challenging to predict the end generalisation performance during the start of training (first 3 days). Although reducing the learning rate can result in immediate performance gains, in the long run the reduced amount of noise injected into the SGD evolution can result in poor generalization, which is in line with recent results, that encourage to train for longer [24], to allow the SGD process to move away from its initialization and enable wider minima. However training for a long time at a set learning rate, before reducing the learning rate, significantly increases the time of training and therefore debugging and development, yielding this approach impractical, for fast development iteration. We also found that training at a too large of a learning rate also does not result in good generalization, therefore further highlighting the importance of choosing a the right initial learning rate. Re-tuning these parameters every time changes to the remaining architectures, or sampling procedure are made, therefore results in a very laborious, time consuming and computational expensive process. However this process is

important to ensure that changes made during model development are improving end generalization performance and are not due to unknowingly correcting for an arbitrarily chosen learning rate.

The proposed method circumvents much of this learning rate schedule tuning. We also found that by reviewing the available theory of SGD dynamics as described in section (2), alongside experimental findings helped with gaining insight and intuition into this complex hyperparameter tuning process.

## 5.1 Limitations and Future work

One clear limitation of AutoLR is that to run the genetic algorithm more than one training run (three in our experiments) needed to be run in parallel resulting in extra and redundant computation. We argue that this can be justified as overall much computation and time is saved by avoiding repeated hyperparameter tuning. Also, clearly the proposed algorithm introduces new hyper-hyper parameters itself. In future we work plan to test our method on a wider range of medical imaging datasets, and show that the new hyperparameters introduced by our method are robust, and do not require any tuning between tasks.

## 5.2 Conclusion

We proposed and evaluated an adaptive learning rate scheduling algorithm combined with adaptive sampling, to avoid having to hand tune learning rate schedules. As shown in section 4 the algorithm enables fast training, and our results indicate that the final generalization performance is reliable and competitive with handcrafted schedules. Our experimental results suggests that our algorithm achieves state of the art performance on the VISCERAL anatomy CT dataset, for the aorta, lung, kidney, rectus abdominis, spleen, sternum, trachea and bladder, on the VISCERAL anatomy benchmark, and beats human inter-annotator agreement scores on the following organs: aorta, lung, kidney, psoas major, rectus abdominis, spleen, sternum, trachea and bladder. These encouraging results pave the way for using CNNs for robust automatic segmentation within clinical practice, such as surgical planning.

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
