# OpenReview forum: "How to control the learning rate of adaptive sampling schemes"
_MIDL.amsterdam/2018/Conference — Submitted to MIDL 2018_

### Review · AnonReviewer4 · 2018-05-03
**Not very original method, good performance, but not so good evaluation, poor clarity of writing**

**Rating:** 2
**Confidence:** 2

**Review:**

The paper presents a way to automatically adapt the learning rate for neural networks. Finding a good learning rate can be very time consuming and the authors present an idea to do this automatically. While I understand the importance of the work, I find the method rather ad hoc and not very clearly presented and evaluated. The authors propose to adapt the learning rate after every \gamma epochs and show two ways to do this: 1. Hand-crafting a scheme to reduce the learning rate with \lambda every \gamma epochs. 2. Trying different learning rates in parallel and after \gamma epochs retrospectively choose the one that performed best (in terms of validation Dice). They present the second method as their proposed method and show that it performs similar to the first method.
I think the quality of the paper is not too high. I am mainly concerned about the novelty and originality of the proposed method. While the proposed approach is probably new, I am in doubt if this method is original enough to write a paper about. In my eyes, the first thing to try when hand-crafting a parameter is too labor intensive, is to brute force the problem by trying different values and pick the one that performs best. What would be very interesting here, is to study what would be a good heuristic to determine how to choose the proposed new learning rates. But the only thing the authors say here is “we found that having one run that explores a higher learning rate, one continuing at the same as the previously learning rate, and one that explores at half the learning rate, worked well”. There is no evaluation or even reasoning for this claim in the paper. Besides this issue, the evaluation is a bit weird. The authors first evaluate the performance (in terms of Dice) on the validation set to compare approach (1) with (2). But if one chooses a parameter based on validation performance, then of course validation performance goes up. Whether this is done manually or automatically doesn’t really matter. What would be more interesting to see, is whether the test performance increases, but this isn’t presented for (1). In my eyes, the most interesting part of the paper is its good performance on the VISCERAL dataset, which is given only little attention.
My last concern is the clarity of the paper. Quite some variables are undefined (or very hard to find the definition of), like l(\theta, x_n), B(theta), v_i, \gamma, w. The experiments are not very properly described (how were the hand-crafted methods made?), the VISCERAL dataset is not described. Also, I fear that the Background section is rather hard to follow because of missing derivations, and a lot of assumptions that are only mentioned, without discussing to what extend these assumptions hold and what this means for the conclusion. While background information is nice, I think the authors try to say too much in too little text.
Pro’s:
-	Interesting and relevant problem is addressed
-	Somewhat new method in presented
-	Method performs well on VISCERAL dataset
-	Paper gives a lot of background information on the problem at hand
Con’s:
-	Not sure whether the presented method is original enough for a paper
-	Unclarity of background, experiments, and data
-	No evaluation of the proposed method (2) and rivaling method (1) on test performance
-	Evaluation on VISCERAL dataset only on 4 images (out of 20)


**Special Issue:**

No

---

### Review · AnonReviewer1 · 2018-05-03
**The proposed method is interesting and intuitive, but the evaluation is fairly weak. The paper mainly shows performance on the validation set (which is what the proposed method was optimised for) and uses some held-out test set only for a small subset of the methods. The dataset is also quite small. The theoretical analysis is interesting, but has a fairly obvious conclusion and isn't really used in the design of the method.**

**Rating:** 2
**Confidence:** 2

**Review:**

Some thoughts about the message and structure of the paper:

* The proposed method looks useful: it is an interesting and intuitive approach to choose a suitable learning rate.

* The Introduction seems to make a self-contradicting argument: In the second paragraph, it suggests to use SGD instead of methods with an adaptive learning rate (e.g., Adagrad, RMSProp, Adam), because recent work [11] shows that hand-tuned SGD can achieve superior results. So far, so good. At the end of the Introduction, however, we are told that the paper proposes an adaptive learning rate for SGD, because hand-tuning is too time-consuming. That doesn't really make sense.

* The paper mixes two ideas: it proposes an adaptive learning rate, but also adds adaptive sampling. This makes the structure of the paper a little bit confusing: is the adaptive learning rate method useful in itself, or whether it only solves problems that are introduced by adaptive sampling? I think it would have been clearer if the authors had focussed on the adaptive learning rate only.

* The theoretical analysis in section 2.1, while interesting, is not really connected with the rest of the paper. The Introduction promises "insights into how to best adapt the learning rate throughout training", but the main conclusion of the theory section seems to be limited to the observation that the performance of the method is influenced by the learning rate, without specifying what this means for the optimisation. There is little connection between equations (1) to (6) and the design of the algorithm.

* The authors could make a stronger argument for why their proposed AutoLR method is really more efficient than training a single network a little bit longer. It seems that it involves a fair amount of extra work, by training not one but at least three networks (and possibly more if there are more hyperparameters to be optimised).


The paper is missing some detail:

* The description of the handcrafted learning rate schedules (end of page 6) and how they were optimised is very brief. How did the handcrafting work? What is the schedule? What do the hyperparameters eta, lambda and gamma refer to? (I assume they are somewhat similar to those for AutoLR, but a closed-form definition would be helpful.) What do the "five weeks of training time using eight NVIDIA Tesla K40s" refer to? Does that also include the AutoLR time?


The evaluation is not very strong:

* The dataset is small, with only 14 scans for training, 2 scans for validation and 4 for testing. It would have made sense to use some form of cross-validation.

* Many of the results are expressed only as the performance on the validation set (Table 1, Figure 2, Figure 3). This is a problem since the proposed method directly optimises for the performance on the validation set, so it is not surprising that it gives good results. For instance, in Figure 3 the proposed AutoLR looks like it is much more stable than the other methods, but for all we know it might have a much more variable performance on the test set.

* The only results on the test set are shown in Table 2 (I assume this is the case; it would be good to specify this in the caption of Table 2). For some reason this table only shows the results for AutoLR + isample with and without post-processing, but not for any of the baseline methods (AutoLR without isample, the hand-tuned methods).

* Since the paper effectively proposes an alternative to AdaGrad and other approaches with an adaptive learning rate, I think this is something that should have been included in the experiments. Does SGD with AutoLR perform better than one of the existing methods?

* From Figure 3b/d, we can observe that the learning rate schedule chosen by AutoLR does not look too complex: it looks very much like exponential decay. What would happen if the handcrafted parameters were chosen to approximate a similar learning rate schedule? Why is gamma set to 400 for the handcrafted schedule and to a much smaller (smoother) value for AutoLR?


Minor comments

* Page 3, 3.2: In what sense is CT data "sparse"?

* Algorithm 1: It would be clearer if the symbols for the initial learning rate was given directly after the term. What does the final eta symbol on the line "Set the optimal learning rate" mean?

* It is not entirely clear to me what the output of the network is. Is it a single label for a patch? Does the network output a patch, as shown in Figure 1a? How are these patch-wise segmentations combined into a segmentation of the full image?

* How large are the 3D images?

* The paragraph before equation (6): The sentence starting with "If we now run SGD" is not entirely clear: if we do that, then what?

* Last paragraph before section 5.1: This sentence is not grammatically correct.

**Special Issue:**

No

---

### Review · AnonReviewer3 · 2018-05-10
**Relevant topic but problem with clarity and evaluation**

**Rating:** 2
**Confidence:** 2

**Review:**

The authors suggest an adaptive learning rate scheduler of deep learning architectures. The paper includes a mathematical justification for the suggested method based on a couple of assumptions (e.g., linearity of covariances). The method is evaluated on segmenting various anatomical structures from 20 CTA scans.

The paper addresses a relevant topic in deep learning and I appreciate the background material to justify the proposed model. However, I have some concerns with the clarity and the evaluation of the proposed method.

- The mathematical justifications are hard to follow without an in-depth coverage of the cited literature and not well connected to the rest of the paper.

- I would also appreciate a comparison against popular learning rate schedules like cyclical learning rates [25] on commonly used benchmark datasets (e.g., CIFAR10). At this point, I am not confident that the results will generalize across datasets.

- The validation and test datasets are small and except for Table 2 the authors do not present results on the test set but only on the validation set that was used to tune the proposed method.

- I would like to see more details on the choice of the hyperparameters (nu, lambda, gamma) of the manually defined schedules.

- The computational cost increase by threefold when using the proposed approach. Are there any ideas to decrease the computational burden of this adaptive learning rate scheduler?

- Will the source code of this method be publicly available?



**Special Issue:**

No

---

### Decision · Program_Chairs · 2018-05-15
**Paper5 Acceptance Decision**

Reject